# The Complexity of FGF23 Effects on Cardiomyocytes in Normal and Uremic Milieu

**DOI:** 10.3390/cells10051266

**Published:** 2021-05-20

**Authors:** Andreja Figurek, Merita Rroji, Goce Spasovski

**Affiliations:** 1Department of Internal Medicine, Medical Faculty, University of Banja Luka, 78000 Banja Luka, Republic of Srpska, Bosnia and Herzegovina; 2Institute of Anatomy, University of Zurich, 8057 Zurich, Switzerland; 3Department of Nephrology, Faculty of Medicine, University of Medicine Tirana, 1001 Tirana, Albania; meritarroji@yahoo.com; 4Department of Nephrology, Medical Faculty, University of Ss. Cyril and Methodius, 1000 Skopje, North Macedonia; spasovski.goce@gmail.com

**Keywords:** FGF23, FGF signaling, klotho, Wnt pathway, cardiomyocyte

## Abstract

Fibroblast growth factor-23 (FGF23) appears to be one of the most promising biomarkers and predictors of cardiovascular risk in patients with heart disease and normal kidney function, but moreover in those with chronic kidney disease (CKD). This review summarizes the current knowledge of FGF23 mechanisms of action in the myocardium in the physiological and pathophysiological state of CKD, as well as its cross-talk to other important signaling pathways in cardiomyocytes. In this regard, current therapeutic possibilities and future perspectives are also discussed.

## 1. FGF23 Signaling in the Physiological Milieu

FGF23 is a phosphaturic hormone primarily secreted by osteocytes to maintain phosphate and mineral homeostasis. It stimulates phosphate excretion, inhibits PTH secretion, and decreases active vitamin D levels [1,2]. FGF23 was portrayed as a crucial player in the pathogenesis of a variety of hypophosphatemic disorders, including autosomal dominant hypophosphatemic rickets, tumor-induced osteomalacia, and X-linked hypophosphatemic rickets [3].

Osteocytes and osteoblasts mainly produce FGF23. However, low FGF23 messenger RNA (mRNA) levels are detected in the brain, thymus, spleen, small intestine, heart, and testis [4].

It belongs to the superfamily of fibroblast growth factors (FGF) that exert pleiotropic effects on a vast range of biological processes, including the development of organogenesis and metabolism, being an essential part of the endocrine subfamily. While not all FGFs express endogenously in the heart of humans and rodents, FGF2, FGF3, FGF8, FGF9, FGF10, FGF16, FGF15/19, FGF21, and FGF23 act on the heart in a paracrine or endocrine manner inducing physiological or pathological pathways in the heart development, health, and disease [5]. While the paracrine FGFs are not secreted and do not exert their biological functions within the same cell, endocrine FGFs acts as endocrine hormones in targeted tissues [6].

The endocrine family consists of a unique structure lacking conserved heparin (HS)-binding domain, which favors their release from the production source, disadvantaging their FGF receptors (FGFR) activation at targeted organs. They overcome this handicap, binding FGFR by the co-receptor klotho. It seems that the Klotho family of membrane proteins may have somehow evolved to increase the affinity of endocrine FGFs to FGFRs at their target organs [7].

Klotho is a membrane protein that divides into three subtypes: α-Klotho, β-Klotho, and γ-Klotho. Although the α- and β-Klotho interact with FGF23 and FGF19/FGF21, respectively, the function of the third member, γ-Klotho, continues to be unclear. FGF23 activates FGFR 1c, –3c, and –4, but not FGFR2, in the presence of cell surface membrane α-Klotho [8].

FGF23 gene encodes 32-kDa glycoprotein of 251 amino acids. The measurement of the plasma half-life of the biologically active form of FGF is reported to be 45–60 min in humans [9]. The biologically active form consists of two parts: N-terminus FGF core homology domain (155 amino acids) that shares homologies with the other FGF family members and a C-terminal domain (72 amino acids), which is essential for interaction with the FGFR-Klotho complex [9,10].

The full-length biologically active FGF23 can be cleaved by furin pro-protein convertase into N- and C-terminal fragments [11]. Besides mediating the binding of FGF23 to a de novo site generated at the composite FGFR1c-Klotho interface, the 72-residue-long C-terminal tail of FGF23 impairs FGF23 signaling by competing with full-length ligand to bind to the binary FGFR-Klotho complex actively suppressing FGF23 activity [12]. Hence, the cleavage of FGF23 appears to be the primary regulator of its biological effects [10,11,12]. Of note, α-Klotho also has two extracellular domains: KL1 (Glu-34 to Phe-506) and KL2 (Leu-515 to Ser-950). The N-terminus of FGF23 and the KL2 domain of α-Klotho interact with FGFRs, where the C-terminus (CTs) of FGF23 binds to a pocket created by the KL1 and KL2 domains, forming the active ternary receptor complex. The binding affinities of the CTs of endocrine FGFs towards Klotho receptors are shown to be 1000–10,000-fold times stronger than binding affinities of their FGF part towards FGFRs. Hence, it’s clear that Klotho proteins function as the primary surface receptors for endocrine FGFs, whereas FGFRs function as a catalytic subunit of the assembled and activated signaling complex. Besides, it was demonstrated that the CTs of FGF23 would contain two tandem repeats where each repeat binds with high affinity to α-Klotho. Engineered FGF23 variants containing each of the two repeats individually or both repeats bind specifically to α-Klotho and stimulate cell signaling to a similar extent. Thus, the complexity of FGF23 regulation and its role in assembling the FGF23/FGFR/α-Klotho signaling complex [13] is going to be revealed.

This canonical FGF23/FGFR/α-Klotho co-dependent signaling limits FGF23 effects to tissues expressing α-Klotho [14]. Membrane-bound Klotho is mainly expressed in kidney tubules, mostly in distal segments, parathyroid glands, choroid plexus, the sinoatrial node and minimally in bones and cartilage. A cardiac phenotype previously described in Klotho deficiency was sinoatrial node dysfunction, which is the only part of the heart by evidence that expresses Klotho endogenously [15]. Besides, there are only limited data that report Klotho proteins in human cardiomyocytes of atria, supporting the hypothesis that the heart could be an organ that produces its own Klotho proteins [16].

Hence, through binding of FGFR and klotho mainly in the kidney and parathyroid glands, FGF23 functions as an endocrine hormone that regulates phosphorus homeostasis. Whereas FGFRs are abundantly expressed throughout the body, tissue expression of a Klotho is tightly regulated, and therefore determines the effect of FGF23 on target organs [17]. In normal physiological conditions, the kidney plays a crucial role in maintaining soluble Klotho homeostasis. Low Klotho mRNA expression in the kidney was found in CKD patients and positively correlated with their estimated glomerular filtration rate [18]. It starts to decline early in CKD and leads to the rise of serum FGF23, PTH, and Pi.

Additionally, circulating concentrations of FGF23 increase progressively as the renal capacity for phosphorus excretion declines reaching serum levels of up to 1000-fold higher than in healthy individuals in dialysis patients [6]. Data from a recent metanalysis showed the estimated absolute difference in mean FGF23 concentrations between the top to bottom third of the FGF23 distributions ranged from 72 RU/mL in general population studies to 433 RU/mL in non-dialyzed CKD studies and to 8644 RU/mL in dialysis population studies (C-terminal studies only) [19].

Apart from the trans-membrane α-klotho, a soluble form of Klotho (sKL) is generated by proteolytic cleavage (called shedding) of a disintegrin and metalloproteinase-domain containing protein 10 (ADAM-10) and a disintegrin and metalloproteinase 17 (ADAM-17). The sKL is shed from the cell surface into the blood, urine and cerebrospinal fluid following proteolytic cleavage exerting its biological effects on distant organs or tissues. The biological functions of sKL in cardiovascular system may be independent of FGFR-1, including modulation of the endothelial nitric oxide synthesis, maintenance of the endothelial integrity, and inhibition of the growth factor 1 signaling [20]. Besides, it was shown as an independent relationship between high plasma sKL levels and a decreased likelihood of cardiovascular disease (CVD) in humans [21]. Moreover, the elevations of sKL can have FGF23-independent cardioprotective effects through alterations of TRPC6 signaling in the myocardium [22].

Thus, Klotho could block or diminish the hypertrophic effect of FGF-23 on the heart, as it does with the other harmful effect of FGF-23 on cardiac function [23,24]. In this line, the hormonal administration of sKL may be an alternative approach to blocking FGF23 for treating FGF23-induced LVH [25]. On the other side elevations of FGF23, however, are also associated with reductions in sKL, which can modify the development of LVH [15].

CVD is the most frequent (39%) cause of mortality in this population of end-stage renal disease (ESRD), whereas the risk of CV mortality in early-stage CKD far exceeds the risk of progressing to dialysis [26]. Apart from the impact in bone metabolism, recent experimental and clinical studies have demonstrated positive relationship between the endocrine-acting FGF23 and greater risks of major CV events and mortality [6,27].

A probable interpretation was proposed by studies one decade ago where elevated FGF23 was linked independently with greater left ventricular mass and a greater prevalence of left ventricular hypertrophy (LVH) [28,29]. According to initial study reports, since α-Klotho is highly expressed in kidneys but not in the heart, it was supposed that FGF23 might indirectly impact CV homeostasis through its canonical renal actions by the bone–renal–cardiac axis created by FGF-23 “on-target” activation of FGFR/α-Klotho complexes in the kidney.

The afferent limb of this bone–renal–cardiac axis consists of aldosterone, ATII, and β-adrenergic pathways that stimulates FGF23 gene expression in bone. Interestingly, the treatment of rodents with Ang II stimulated the expression of FGF-23 message in bone, the physiological site of FGF-23 production [30,31] and the ectopic expression of FGF23 in the heart, a tissue that does not normally express FGF-23 or α-Klotho in physiological amounts [30]. In addition, Ang II administration increases circulating FGF-23 levels in animal models [32]. Hence, the increased FGF-23 might result from direct effects to activate AT1 receptors in osteoblasts or secondary to Ang II stimulation of aldosterone and TNF-α production, both of which can increase FGF-23 or suppress the Klotho expression in the kidney, leading to an end-organ resistance and secondary elevations in FGF-23 [32]. Since FGF-23 levels are higher in patients with heart failure not treated with angiotensin-converting enzyme inhibitors (ACEi) and patients in the top tercile of elevated serum FGF-23 show a lower risk of adverse events after treatment with ACEi points to the clinical importance of cross-talk between RAAS and FGF-23 [33]. The RAAS signaling plays an essential role in FGF23 induction in the heart, and both signaling pathways synergistically contribute to cardiac remodeling [34].

While FGF23 itself did not directly influence the expression levels of any fibrosis-related mRNAs, FGF23 was shown to enhance the effect of TGF-β1 on raising the expression levels of α-smooth muscle actin (α-SMA) mRNA through the inhibition of FGFR1 or the knockdown of FGFR1 in fibroblasts [35]. Thus, FGF23 synergistically promoted the activation of fibroblasts with TGF-β1, transforming fibroblasts into myofibroblasts via FGFR1. An animal model has identified FGF23 as a paracrine factor secreted from cardiomyocytes to promote cardiac fibrosis under a condition in which TGF-β1 was activated [35,36]. Like RAAS, the sympathetic nervous system (SNS) stimulates FGF-23 production in bone through β-adrenergic signaling pathways [36,37].

The efferent limb consists of FGF23 activation of FGFRs/α-Klotho in renal tubules through at least four mechanisms: induction of hypertension by enhancing Na reabsorption and suppressing ACE2 [35,38]. The other possible mechanisms investigate the activation of RAAS through either suppression of 1.25 (OH2)D, which, in turn, increase the renin expression and reduction of ACE2 expression [39]; ectopic expression of FGF23 and α-Klotho in the stressed myocardium and not in the normal heart [40]; and suppression of FGF23 of α-Klotho [41] that may lead to LVH through loss of the sKL cardioprotective effec [42].

The uremic environment seems to prime the cardiomyocytes to FGF23 signaling since the left ventricular function was reported normal in patients suffering from increased FGF23 due to genetic causes and an animal model of x-linked hypophosphatemia [43,44]. Although the genetic disease is associated with disturbed FGF23 homeostasis, the plasma levels of FGF23 do not reach the extreme levels as seen in CKD patients [45].

Besides the involvement of the RAAS [38], experimental models showed that FGF23 affects the immune system triggering the liver to produce inflammatory cytokines such as IL-6 and CRP via FGFR4 [46].

FGF-23 is proposed to impair host responses to infection and impair innate immune responses [47]. Moreover, CKD is known to be a state of inflammation and impairment of the immune system [45]. Thus, an interplay between the systemic regulatory inflammation and local factors regulating FGF23 was noted, proposing that a positive feedback loop may exist where FGF23 promotes the secretion of pro-inflammatory cytokines inducing higher FGF23 secretion [48].

In 2011, Faul and colleagues [29] initially tested the hypothesis about an independent connection between elevated FGF23 levels and grown risk of LVH in patients with CKD. The hypertrophic effects of FGF23 were mediated by activation of cardiac FGF receptor 4 (FGFR4), resulting in the recruitment and phosphorylation of phospholipase Cγ (PLCγ) and a subsequent activation of the calcineurin/nuclear factor of activated T cells (NFAT) signaling cascade, not requiring klotho as coreceptor [49]. NFAT is a potent inducer of ventricular remodeling in response to different pathogenic stimuli [29,49]. Besides, it was shown that treatment with PLCγ inhibitor, U73122, attenuated FGF23-mediated hypertrophy but had no effect on FGF2-treated cells. The pro-hypertrophic effects of the prototypical paracrine FGF family member, FGF2, were blocked by inhibitors of the Ras (a small GTPase)/mitogen-activated protein kinase (Ras/MAPK) cascade. These effects on cardiomyopathy were independent of blood pressure levels. Later on, it was shown that FGF23/FGFR4 signaling reversibly increases cardiac contractility and hypertrophy in vitro and in vivo and that an isoform-specific FGFR4 blocking antibody is reversing FGF23-mediated hypertrophic growth of isolated cardiac myocytes and established LVH in the animal model of CKD with high FGF23 levels [50]. It was recently found that the conditional deletion of FGFR4 in the myocardium prevented rFGF23-induced LVH in mice, establishing the direct cardiotoxicity role of FGF23 through an activation of FGFR4. Another exciting finding reported by these authors was sKlotho administration’s ability to prevent FGF23-induced LVH and the effects of sKlotho to attenuate FGF23/FGFR/PLCγ signaling both in vivo and in vitro [25]. Furthermore, it was shown that FGF23 increases intracellular calcium levels in cardiac myocytes in vitro and promotes contractility of murine cardiac myocytes and ventricular muscle strips ex vivo independently from α-klotho.

## 2. FGF23 Signaling in the Uremic Milieu

### 2.1. Cardiac Hypertrophy

As patients with CKD are at higher CV risk compared to the healthy population, it is important to understand underlying pathophysiological mechanisms and, subsequently, reduce the mentioned risk. Thus, a reliable biomarker that could estimate CV risk and mortality should be established. Hence, FGF23 was found as a sensitive biomarker elevated in early CKD [51] suitable for timely estimation of CV risk and therapeutical response in order to possibly prevent CV events and decrease mortality in those patients. Nowadays, many clinical trials reported that FGF23 is very close to being introduced into the clinical routine.

Although primarily secreted by osteocytes in bones, CKD patients have higher circulated FGF23 levels mainly due to the declining glomerular filtration and excretion by the kidneys [52,53]. In addition, increased FGF23 production in CKD (due to the impact of secondary hyperparathyroidism, treatment with active vitamin D metabolites, etc.) may contribute to the increased FGF23 plasma levels.

The pathological process starts with binding the FGF23 to its receptor in cardiomyocyte, leading to cardiac remodeling and hypertrophy that could be clinically detected. Therefore and not surprisingly, the prevalence of LVH in the general population is about 15–20% [54], whereas in the CKD population, it’s more than 70% [55].

Although experimental data in mice indicated the relative expression of FGFR3 and FGFR1 in isolated adult ventricular cardiomyocytes is higher compared to FGFR4 [56], the major functional significance in cardiac hypertrophy development seems to belong to FGFR4. Even though previously αKlotho was thought not to be expressed in cardiomyocytes, there are some experimental data indicating still very low expression [56], detected through a very high threshold cycle for αKlotho mRNA levels in the PCR analysis and with no confirmation with in situ hybridization and immunohistochemistry. This would support the fact that the role of αKlotho is not crucial in FGF23 signaling in cardiomyocytes.

It is important to underline that high FGF23 levels induce upregulation of FGFR4 expression, indicating the induction of cardiac hypertrophy via FGFR4 [57]. On the other hand, experimental studies showed that, even on a high phosphate diet, cardiac hypertrophy does not occur when FGFR4 is blocked [50]. Moreover, in experimental settings, cardiac hypertrophy is shown to be reversible [50]. Additionally, even Klotho deficiency, which also exists in CKD, contributes to cardiac hypertrophy development [58]. 

A generally accepted understanding is that FGF23 binds to the FGFR4 and exerts its mechanism, which is Klotho independent (Figure 1). This binding results in FGFR4 auto-phosphorylation and subsequent binding of phospholipase C y (PLCy) to the phosphorylated FGFR4 sequence activating PLCy [59]. Activated PLCy results in calcineurin activation and nuclear factor of activated T-cells (NFAT) dephosphorylation, which is then translocated to the nucleus with ensuing transcription of pro-hypertrophic genes (brain natriuretic peptide (BNP), regulator of calcineurin 1 (Rcan1), β-myosin heavy chain (β-MHC), atrial natriuretic peptide (ANP), etc.) [25,60,61,62,63]. Of note, ANP and BNP are well-defined markers of cardiac hypertrophy [64].

Acute treatment with FGF23 increased the early growth response 1 (EGF1) gene and the hypertrophy-associated genes ANP and BNP after 24 h, whereas after 48 h the expression of β-MHC and skeletal muscle α-actin (SkAct) occurred with dose-dependent increase in cell size [56]. Furthermore, it was shown that FGF23 treatment increased ERK phosphorylation 15 min upon the treatment [56].

Experimental studies in rats showed that cardiomyocyte size, synthesis of cardiac FGF23 and FGFR4 expression negatively correlated with phosphorylated NFAT [65], confirming the importance of FGF23/FGFR4/calcineurin/NFAT signaling in the myocardium. 

Further studies revealed the importance of calcium (Ca^2+^) homeostasis dysregulation and the cross-talk to FGF23 signaling in cardiac hypertrophy. Acute FGF23 treatment increased intracellular Ca^2+^ in ventricular myocytes by promoting Ca^2+^ entry via L-type Ca^2+^ channels (Figure 1). Thus, the cardiac contractility is altered by increasing force and rate of force development, indicating that long-term disruption of Ca^2+^ homeostasis in cardiomyocytes might cause contractile dysfunction, heart remodeling and, finally, cardiac hypertrophy [56].

Depolarization of the cardiomyocyte membrane action potential opens L-type calcium channels at the sarcolemma allowing the Ca^2+^ influx into the cytoplasm [66] that activates ryanodine receptors (RyRs), triggering a greater amount of Ca^2+^ release from the sarcoplasmic reticulum into the cytoplasm [67].

After an increase in the intracellular Ca^2+^ concentration, Ca^2+^ binds to troponin C at the myofilaments, resulting in cardiomyocyte contraction [67]. To return to a previous intracellular Ca^2+^ concentration in order to conduct relaxation, cytosolic Ca^2+^ is either pumped to the sarcoplasmic reticulum (92% of cytosolic Ca^2+^) by the action of the SR–Ca^2+^–adenosine triphosphatase 2a (SERCA) or is extruded to the extracellular space by the Na^+^–Ca^2+^ exchanger (NCX) (7% of cytosolic Ca^2+^) [67]. Calmodulin kinase II (CaMKII) and cAMP-dependent protein kinase (PKA) are involved in the opening of RyRs and they can also phosphorylate phospholamban (PLB), a sarcoplasmic reticulum membrane protein that regulates SERCA activity [63,68].

Experimental studies indicated FGF23 is involved in this Ca^2+^ handling increasing phosphorylation of regulator proteins, such as CaMKII, inducing cardiomyocytes to develop a cellular phenotype related to contractile dysfunction and predisposition to arrhythmias [63,69,70,71].

FGF23 may induce pro-arrhythmogenic Ca^2+^ events that may result from RyRs hyperactivity and a reduction in SERCA activity with subsequent increase in cytosolic Ca^2+^ levels, further leading to impaired cardiomyocyte contraction, represented by less cell shortening [23,70]. Of note, the treatment of mice recombinant Klotho prevented cardiac dysfunction and pro-arrhythmogenic Ca^2+^ release events [23].

In vitro studies showed that both FGF23- and ATII-induced Ca^2+^ release from the intracellular stores have been associated with cardiomyocyte hypertrophy [72]. Interestingly, FGF23 enhances angiotensin II (ATII) expression and secretion in cardiomyocytes, and angiotensin II receptor type 1 (AT1R) antagonist losartan attenuates FGF23-induced effects on cardiomyocyte hypertrophy and Ca^2+^ signaling [72]. 

Besides the connection to Ca^2+^ signaling and RAAS in developing myocardial hypertrophy, studies also pointed out the importance of Wnt signaling and the cross-talk to FGF23 in this pathology. Experimental studies in mice indicated that cardiac remodeling was related to a reduction of Klotho and activation of the Wnt-β-catenin pathway and renin-angiotensin system [73].

The heart reactivates several signaling pathways on pathological stress that traditionally were thought to be operational only in the developing heart and one of these important pathways is the Wnt signaling pathway (Wingless-related integration site), known to be important in embryonical development and tissue renewal [74]. Wnt pathway has several sub-pathways defined either as non-canonical (Wnt/Ca^2+^ pathway and non-canonical Wnt planar cell polarity) or as canonical Wnt-pathway (β-catenin-T-cell factor/lymphoid enhancer-binding factor (TCF/LEF)) [75].

Wnt activates the non-canonical pathway including the axis of phospholipase C, which, in turn, leads to calcium (Ca^2+^) release within the cell with subsequently induced Ca^2+^/calmodulin kinase (CaMK)II and calcineurin activation [74].

In the absence of Wnt ligand, the destruction complex containing Adenomatous Polyposis Coli (APC), Axin, and GSK3β and glycogen synthase kinase 3β (GSK3β) phosphorylate β-catenin, which is then recognized by ubiquitin ligase β-Trcp and degraded by ubiquitin proteasome pathway [76,77], takes place. An important role in the regulation of Wnt signaling is of Yes-associated protein (YAP)/transcriptional coactivator with PDZ-binding domain (TAZ), as the prime mediators of the Hippo pathway and as an important regulator of cellular proliferation and differentiation [76]. They are involved in destruction complex formation, and during Wnt activation, YAP/TAZ accumulates in the nucleus, while when Wnt signaling is inactive, YAP/TAZ is essential for β-Trcp recruitment to the complex and β-catenin inactivation [76]. It has been proposed that the Hippo pathway restricts Wnt signaling via interaction between TAZ and DVL [76].

When Wnt ligands bind to the Frizzled family receptors and display an interaction with co-receptors lipoprotein-receptor-related protein 5 (LRP5) and LRP6, canonical Wnt pathway activates and stimulates the sequestration of Axin protein by the Disheveled protein. This results in prevention of the destruction complex formation that is necessary for the β-catenin degradation [78]. Canonical Wnt pathway activation leads to inhibition of GSK3β via intracellular binding of Axin, thereby preventing β-catenin phosphorylation. [74] Hence, β-catenin is not subject to phosphorylation, and once stabilized, it is translocated into the nucleus, where the transcription of the Wnt target genes is activated via the interaction with the transcription factors TCF/LEF [78,79]. When β-catenin is present, it displaces Groucho/TLE (the human transducing-like Enhancer of split) from TCF/LEF by binding to the low-affinity binding site on LEF-1 that includes sequences of N-terminal to the DNA-binding domain, and that overlaps the Groucho/TLE-binding site [80] (Figure 1). In the absence of nuclear β-catenin, TCF/LEFs act as transcriptional repressor by binding to Groucho/TLE proteins.

Wnt-dependent activation of the canonical Wnt-pathway is blocked by the soluble frizzled-related proteins (sFRPs) that are secreted from autologous bone marrow-derived mononuclear cells acting as decoy receptors for Wnt ligands, inhibiting both canonical and non-canonical Wnt signaling. Thus, the ratio of cytoplasmatic and nuclear β-catenin is altered, the nuclear amount of β-catenin is lower and subsequently β-catenin-dependent transcription is reduced, which results in the left ventricular remodeling attenuation as shown in experimental studies [74]. Important inhibitors of canonical Wnt signaling are Dickkopf-related protein 1 (DKK1) and sclerostin. DKK1 forms a tertiary complex with LRP5/6 and the cell surface co-receptor Kremen-1 with the subsequent internalization of the receptor complex, whereas sclerostin binds to LRP5/6 and prevents binding of Wnt ligands to Frizzled-LRP5/6 receptor complex [81]. Experimental studies indicate that increased DKK1 levels are associated with the alleviation of cardiac hypertrophy [82]. Targeting other parts of Wnt signaling, e.g., Disheveled depletion showed to attenuate left ventricular remodeling [74].

Current experimental findings support the cardioprotective role of Klotho. While ATII promotes Wnt/β-catenin pathway activation, Klotho, on the other hand, reduces the expression of AT1R and reduces the angiotensin II-induced hypertrophy of neonatal cardiomyocytes [83].

Data from the clinical studies support the important role of FGF23/Klotho/Wnt pathway cross-talk, where it was shown that in patients on hemodialysis elevated serum FGF23 with lower sKlotho and sclerostin, may act as a predictor of CV complications (LVH, acute coronary syndrome, arrhythmias, etc.) [84].

Increased renin production results in increased conversion of angiotensinogen to angiotensin I, which is further metabolized to ATII by ACE [85]. By activation of angiotensin II receptor type 1 (AT1R), ATII causes peripheral vasoconstriction, activation of sympathetic nervous system, and secretion of aldosterone from the adrenal glands. This, in turn, may increase sodium and water reabsorption in the distal kidney tubules, leading to an increased blood pressure [85,86]. Here, it is important to underline that in the development of cardiac hypertrophy in CKD, both systemic and local RAAS play a role. 

An important part of the RAAS is ACE2 which degrades angiotensin I and II into angiotensin 1–9 and 1–7, that have vasodilatory and hypotensive effect [6]. FGF23 stimulates RAAS directly by inhibiting ACE2 [41]. In addition, RAAS activation induces FGF23 synthesis [37].

Moreover, it has been demonstrated that ATII and aldosterone increase serum FGF23 levels and that ATII increases the expression of cardiac-specific FGF23 [32]. Additionally, as FGF23 suppresses 1.25(OH_2_)D_3_ renal metabolism and knowing that 1.25(OH_2_)D_3_ suppresses renin and RAAS activation, it is likely that FGF23 exerts indirect action on RAAS activity [1,6].

### 2.2. Cardiac Fibrosis

Fibrosis is a process characterized by the production and deposition of an extracellular matrix, that can occur in any tissue or organ. Although it might be protective to a certain extent (for instance, after myocardial infarction when replacing the dead cells or during a scar-formation), it expands in the tissues increasing the mechanical pressure on it and devastating normal functional tissue.

Besides the role in cardiac hypertrophy, current findings indicate that FGF23 is involved in the pathophysiological signaling pathways that result in the development of cardiac fibrosis. Experimental studies in rats pointed out that left ventricular fibrosis did not correlate with FGFR4 and NFAT activation, indicating that the fibrotic process was not mediated through the activation of FGF23/FGFR4/calcineurin/NFAT pathway [85]. This finding underlines the significance of the other signaling pathways, that are involved in the cardiac fibrosis occurrence, primarily FGF23/Klotho, TGFβ-, Wnt-signaling, and local RAAS.

Unlike cardiac hypertrophy, where the main pathological place occurs in cardiomyocytes, in cardiac fibrosis in addition to cardiomyocytes, the role of fibroblasts appears as crucial. In addition, although FGF23 exerts its mechanism of action by binding to the FGFR4 in cardiomyocytes, in cardiac fibroblasts, FGF23 additionally regulates TGFβ1-signaling by acting on its fibroblast growth factor receptor-1 (FGFR1) expressed in fibroblasts [34]. Experimental studies indicated that FGF23 promotes proliferation of cardiac fibroblasts and myocardial fibrosis, both associated with the upregulation of active β-catenin and TGF-β [87].

Cardiac damage may lead to a local synthesis of FGF23 in the heart with fibroblasts as main producers increasing their proliferation, migration, and the expression of β-catenin, TGFβ1, fibronectin, and collagen I [87,88].

The cross-talk between TGFβ- and Wnt-signaling is bidirectional. TGFβ1-signaling can induce the expression of β-catenin superfamily members and β-catenin makes a complex with Smads in the nucleus, which stimulates the transcription of profibrotic genes [89]. TGFβ is able to deactivate glycogen synthase kinase 3β (GSK3β) directly and to activate Wnt/β-catenin signaling through the production of Wnt proteins [90]. Activated Wnt/β-catenin also stabilizes TGFβ/Smad response, making this co-activation of the two pathways effective in triggering the fibrotic response (Figure 2) [90]. Detailed cross-talk between TGF-β, Wnt, and YAP/TAZ is beyond the scope of this review, but is nicely summarized in the review of Piersma et al. [91].

Conversely, Klotho seems to have a protective role as the upregulation of endogenous Klotho-inhibited Wnt/β-catenin signaling [92]. Another encouraging fact is that secreted Klotho was able to inhibit TGFβ1 signaling by binding to the type-II TGFβ receptor inhibiting TGFβ1 binding to cell surface receptors, thus suppressing the renal fibrosis development [93]. On the other hand, loss of endogenous Klotho in cardiomyocytes, as observed in CKD, intensifies TGFβ1 signaling, and Wnt-signaling-mediated cardiac fibrosis occurrence [92].

Local RAAS is involved too. Besides expressing angiotensin II-type 1 and 2 receptors (AT1R and AT2R), cardiac myocytes and fibroblasts also express angiotensinogen, with mRNA levels found to be higher in cardiomyocytes than in fibroblasts [35]. FGF23 treatment stimulated pro-hypertrophic and pro-fibrotic molecules (TGFβ, collagen I, connective tissue growth factor (CTGF), and endothelin-1), known to be induced by ATII and/or aldosterone, in cardiomyocytes and fibroblasts [35]. On the other hand, ATII and aldosterone upregulated the expression of endogenous FGF23 in cardiomyocytes and induced hypertrophic cell growth, βMHC, ANP, and BNP [6]. In this regard, cardiac FGF23 significantly correlated with endogenous angiotensinogen mRNA expression, suggesting a direct Connection between FGF23, local RAAS, and the progression of left ventricular fibrosis [85].

## 3. Future Perspectives

In CKD, the main pathophysiological mechanism that connects the kidney with bone and cardiovascular system is defined as chronic kidney disease—mineral and bone disorder (CKD-MBD) with FGF23 playing the central role. As confirmed in the current basic and clinical research, the elevated FGF23 seems to play an important role in cardiac pathophysiology. Compensatory elevated FGF23 levels might be beneficial in acute settings and the optimal FGF23 levels are not known [49]. Thus, the question when to intervene with lowering FGF23 levels is not completely answered and direct FGF23 blocking might be problematic for a longer period. Additionally, there are in vivo experimental data suggesting that pharmacological blockade of FGF23 using pan-blocking antibody did not prevent the occurrence of LVH and, instead, caused increased aortic calcification due to hyperphosphatemia and subsequently increased mortality of animals. [94] Similarly, the effect of a pan-FGFR inhibitor resulted in an increased FGF23 and phosphate levels with subsequent multiorgan and multifocal soft tissue mineralization [95]. Therefore, burosumab, which is a fully humanized monoclonal IgG1 antibody, might be used in the cases of primary FGF23 excess, such as X-linked hypophosphatemia with the clinical picture of phosphate losing-osteomalacia. Conversely, the use in conditions with a secondary FGF23 excess as seen in CKD, seems not to be justified at present [49].

In vitro and in vivo studies indicated that FGF23-induced cardiac hypertrophy is reversible by blocking FGFR4 with an isoform-specific FGFR4 blocking antibody, whereas myocardial fibrosis did not completely recover [50], suggesting that in cardiac fibrosis there might be other signaling pathways besides FGFR4/calcineurin/NFAT signaling. 

Many studies conducted so far have tried to develop a way to reduce or to stop cardiac hypertrophy and fibrosis development. Bearing in mind the role of FGF23/FGFR4/calcineurin/NFAT pathway, as well as the importance of local RAAS activation, the possible effect of calcineurin inhibitor cyclosporine A (CsA), angiotensin receptor blocker losartan (Los), and steroidal mineralocorticoid receptor (MR) antagonist spironolactone was investigated. [85] Losartan-induced blockade of AT_1_R and spironolactone-induced blockade of MR led to prevention of FGF23-mediated cardiomyocyte hypertrophy to the same degree as with CsA [85].

It has been demonstrated that by binding to the AT1R, ATII induces extracellular matrix (ECM) proteins, TGFβ, and endothelin-1 expression with subsequent binding of TGFβ to its receptor with further contribution to a profibrotic response [96]. FGF23 enhanced Tgfb and Ctgf mRNA levels, with the trend of reduction upon pharmacological blockade of AT1R and MR, and FGF23-stimulated Col1 expression was reduced upon CsA, Los, and spironolactone treatment [96].

There are a couple of other studies that further indicated a cross-talk between FGF23 signaling, RAAS, and Stat3 and Smad pathways in developing cardiac hypertrophy and fibrosis. However, a detailed investigation on the exact role of specific signaling mediators and transcription factors is needed [97,98,99,100,101].

Potential role of Wnt inhibitors in the pathophysiology of cardiac hypertrophy and fibrosis, such as sclerostin, needs to be further elucidated, as it is known from clinical studies reporting contradictory conclusions about the prediction of CV risk with sclerostin in patients with CKD [102,103]. This question might help for better understanding whether inhibition of Wnt/β-catenin signaling pathway is beneficial in order to reduce the CV risk by controlling cardiac hypertrophy and fibrosis development.

Clinical studies support the beneficial effect of RAAS blockade (e.g., the use of ACE inhibitors) in reducing CV risk by lowering FGF23 levels [33,104]. Additionally, indirect lowering of FGF23 showed to be beneficial in CKD patients. In the Evaluation of Cinacalcet HCl therapy to Lower Cardiovascular Events (EVOLVE) trial, where calcimimetic cinacalcet was used for the treatment of secondary hyperparathyroidism in CKD, FGF23 levels were found reduced, and moreover, the decrease in FGF23 levels >30% resulted in significant reduction in CV mortality [105].

Taken together, current findings from basic and clinical research indicate that FGF23 effects on cardiac hypertrophy and cardiac fibrosis development, could be controlled by blocking the activation of FGF23-mediated cross-talk mechanisms in a safe way, rather than direct blocking of FGF23. Indirect lowering of FGF23 levels by reducing phosphate load [106] or by treatment with calcimimetic cinacalcet and low-dose calcitriol analogs [107] seems to have beneficial effect as well. Although the effect of activated vitamin D results in LVH attenuation in both animal studies [108] and in hemodialysis patients [109], being partially explained due to its parathyroid hormone-lowering effect, there is still a concern regarding the use of calcitriol analogs alone, since the activated vitamin D is one of the most potent stimulators of FGF23 production in bone increasing the FGF23 levels [110]. Other possibilities could be using sKlotho or L-type calcium channel blocker in order to be able to control this pathophysiological process.

Finally, inactivation of the FGFR4/calcineurin/NFAT pathway, as well as other important signaling pathways that are activated by FGF23, requires more translational and clinical research in order to draw a conclusion about the efficacy and safety of their potential use in the clinical practice. This is particularly important in the setting of whole complexity of CKD pathophysiology with various factors contributing to the heart pathology development: anemia, chronic inflammation, oxidative stress, and uremic toxins, in particular, those that are not sufficiently removed by dialysis, such as indoxyl sulfate and p-cresyl sulfate that shows direct cardiotoxicity [111].

## Figures and Tables

**Figure 1 cells-10-01266-f001:**
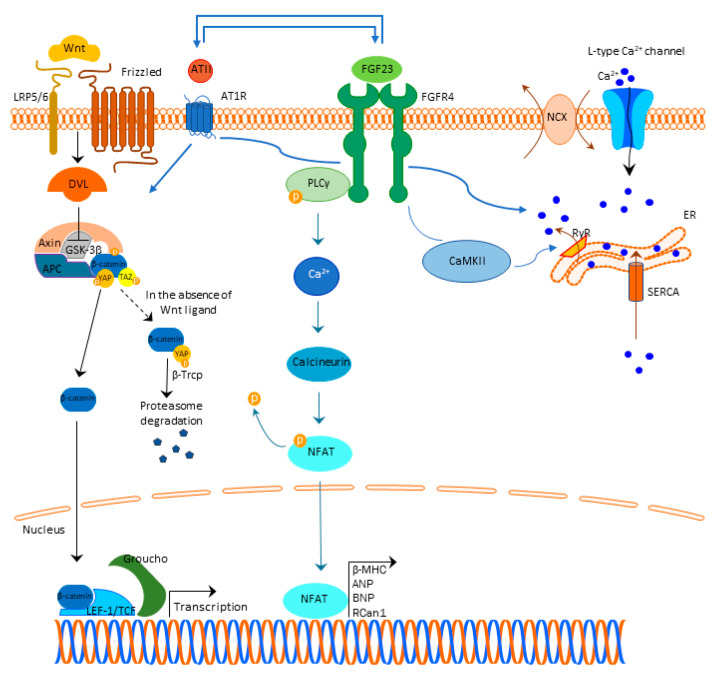
FGF23 signaling and cardiac hypertrophy. Abbreviations: FGF23-fibroblast growth factor 23, FGFR4—fibroblast growth factor receptor 4, PLCy—phospholipase C y, Ca^2+^—calcium, NFAT—nuclear factor of activated T-cells, β-MHC—β-myosin heavy chain (β-MHC), ANP—atrial natriuretic peptide, BNP—brain natriuretic peptide, Rcan1—regulator of calcineurin 1, LRP 5/6—lipoprotein-receptor-related proteins 5 and 6, DLV—Disheveled protein, APC—Adenomatous Polyposis Coli, GSK3β—glycogen synthase kinase 3β, YAP—Yes-associated protein, TAZ—transcriptional coactivator with PDZ-binding domain, β-Trcp—beta-transducin repeat containing E3 ubiquitin protein ligase, LEF/TCF—lymphoid enhancer-binding factor/β-catenin-T-cell factor, ATII—angiotensin II, AT1R—angiotensin II receptor type 1, NCX—sodium-calcium exchanger, ER—endoplasmic reticulum, RyR—ryanodine receptor, SERCA—sarcoendoplasmic reticulum Ca^2+^ ATP-ase, CaMKII—Ca^2+^/calmodulin kinase II.

**Figure 2 cells-10-01266-f002:**
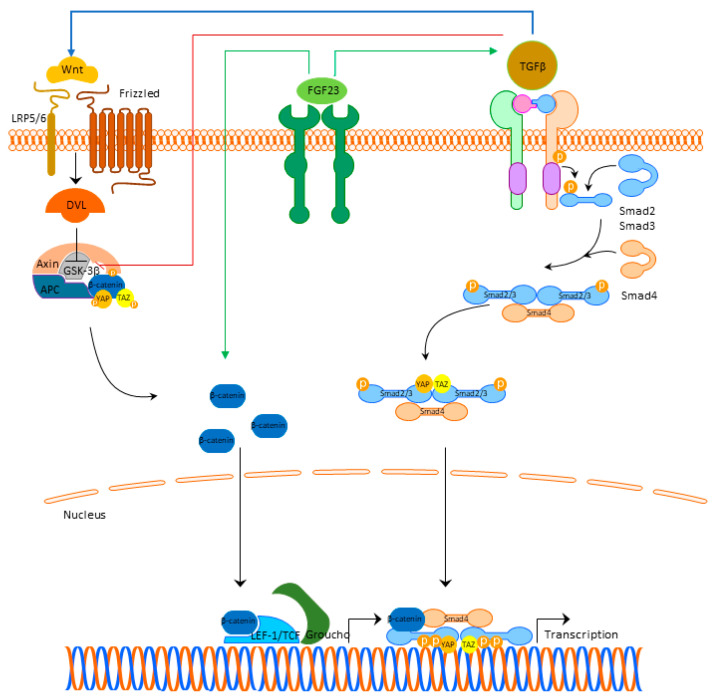
FGF23 signaling and cardiac fibrosis. Abbreviations: FGF23—fibroblast growth factor 23, FGFR4—fibroblast growth factor receptor 4, LRP 5/6—lipoprotein-receptor-related proteins 5 and 6, DLV—Disheveled protein, APC—Adenomatous Polyposis Coli, GSK3β—glycogen synthase kinase 3β, YAP—Yes-associated protein, TAZ—transcriptional coactivator with PDZ-binding domain, LEF/TCF—lymphoid enhancer-binding factor/β-catenin-T-cell factor, TGFβ—transforming growth factor beta.

## Data Availability

Not applicable.

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
