# Peer review of "The Complexity of FGF23 Effects on Cardiomyocytes in Normal and Uremic Milieu"

_cells, 2021, doi:10.3390/cells10051266_

Round 1

Reviewer 1 Report

The authors have summarized the effects of FGF23 on the myocardium in uremia and in normal physiology. While these conditions form two clear sections of this review, the sections themselves are a bit jumbled at times. Overall, most of the important effects and concepts are addressed in this manuscript, but the following paragraphs might require some work:

  • On FGF23-Klotho interaction: it would be appropriate to include more data on the binding sites and affinity (e.g. Suzuki et al., PMID 33257569).
  • Soluble Klotho is principally a kidney-derived protein of an approximate size of 130 kDa rather than the 70 kDa form the authors describe.
  • There is quite a number of high-quality studies that indicate that Klotho (overexpression or protein supplementation) prevent myocardial hypertrophy and fibrosis. This is mentioned only in a cursory manner. While this may be outside the scope of this review, I would argue that it is always relevant to ask the question when discussing these experimental studies whether LVH occurred because of an increase of FGF23 or because of a down-regulation of Klotho, and where these effects might intersect.
  • There is very little solid evidence for ectopic expression of Klotho in the heart, aside from the sinoatrial node.
  • The entirety of paragraph 1.2 on Wnt signaling does not fit. It only explains the basics of Wnt signaling and does not tie in to FGF23 signaling in any discernible fashion.
  • Figure 1 is a bit unclear.
    • FGF23 forms a ternary complex with FGFR1c and Klotho. FGF23 has a very low affinity for FGFR1c when not bound to Klotho, which is what Figure 1 depicts.
    • Figure 1 indicates that Klotho inhibits AT2R. While evidence exists that Klotho leads to a reduction in the expression of various RAAS factors, I am not aware of any evidence for a direct inhibitory effect of Klotho on AT2R.
    • Klotho does, however, directly inhibit Wnt signaling, which is not depicted in the Figure (and which is only addressed in cursory fashion in the text).
  • In Figure 2, it is not quite clear where Klotho targets Wnt and TGFb signaling.

On the whole of the manuscript:

  • The authors aim to discuss FGF23 effects under “physiological” conditions and then in uremia.
    • Uremia itself, however, is hardly discussed as a factor. Do uremic toxins play or role? Or do the authors mean to implicate the several thousandfold increase in FGF23 levels, which may compensate for a lower binding affinity to other receptors, allowing activation of pathways that are not active in normal physiology? This point is not discussed clearly enough.
    • It is also unclear why the uremia section of the review details RAAS activation and TGFb signaling, while Wnt signaling is discussed in both sections.
  • The entire review would benefit from proof-reading by a native English speaker – while the language is quite good, there are minor errors throughout the manuscript.

Reviewer 2 Report

Regarding: The complexity of FGF23 effects on cardiomyocytes in normal and uremic milieu by Andreja Figurek Merita Rroji, and Goce Spasovski.

The review nicely summarizes the current knowledge on mechanisms of FGF23 in general physiology and its detrimental effects on the heart in pathophysiological conditions. The review is up to date including the key studies in the field. It is well written.

I do have minor remarks to some statements, the figures and references. Please see below for details. I have placed the comments in chronological order.

Line 24: Please change references 1 & 2 to more suitable ones.

Line 27: Please change reference 3 to a more suitable one.

Line 52: Please note that it is the measured half-life of tumoral FGF23. The “46-58 minutes” is misleading as the 46 minutes refers to the mean clearance of intact & c-terminal FGF23 and the 58 minutes refers to the mean clearance of intact FGF23 in the study. There is considerable variation in the estimation of FGF23 clearance between the patients. Takeuchi Y et al. reported a half-life of tumoral FGF23 to 22 minutes in one case report (Takeuchi Y et al, J Clin Endocrinol Metab 2004). Experimental studies have reported FGF23 half-life of 22 minutes in the mouse (Christov et al, Kidney Int 2013) and 4 minutes in the rat (Mace et al, Kidney Int 2015). In my point of view, the exact half-life of FGF23 in humans is not completely clarified. However, it is most likely shorter than 58 minutes.

Line 58: Regarding reference 11. Is it not reference 15 (Tagliabracci et al, PNAS 2014) you would put here?

Line 58-59. It is a very elegant study by Goetz et al. However, it is still not clear the relevance of c-terminal FGF23 as an antagonist for intact FGF23 signaling in humans. Maybe you could elaborate the statement.

Line 70: Regarding reference 15. Please se above.

Line 73: A reference seems to be missing.

Line 95: You could consider to cite the key studies by Gutiérrez OM et al, Circulation 2009 and Faul C et al, JCI 2011.

Lines 100-107. I cannot find a reference to the nice studies done by Professor Erben and Andrukhova. I suggest that you consider to include references to these studies.

Lines 108-130.Comment to the section: it seems that something in the uremic environment may prime the cardiomyocytes to FGF23 signaling since left ventricular function was found normal in patients suffering from increased FGF23 by genetic causes and in an animal model of x-linked  hypophosphatemia (Hernandez-Frias O et al, Pediatr Nephrol 2019 & Pastor-Arroyo, E. M et al, Kidney Int 2018). Would you comment on this?  

Line 147 “Wnt leads to inhibition of GSK3β” – do you mean Wnt ligands, Wnt activation (?). There seems to be word missing from the sentence.

Line 167. I do agree with the statement, underlining the key role of the kidney in the regulation of plasma levels of FGF23. However, it needs a reference. Still, the impact of secondary hyperparathyroidism and treatment with active vitamin D metabolites (as well as other factors) are important stimulators of FGF23 production in CKD, contributing to the increase plasma levels of FGF23. The impact of these factors may differ across the different stages of kidney failure. Also, it is not known the impact of the different types of renal osteodystrophy has on production of FGF23 in bone. I do know that it is beyond the scope of the review to discus this matter. However, you could add a short sentence saying that increased FGF23 production in CKD may also contribute to the increased plasma levels. So, your text not only states decreased renal clearance as the sole factor.

Line 176. In the study by Touchberry, CD et al, they report a a very high threshold cycle for αKlotho mRNA levels in the PCR analysis. One may suspect it is not physiological relevant. Also, the findings were not confirmed with in situ hybridization and immunohistochemistry. Any comments on this?  

Line 250-251. SFRPs function as decoy receptors for Wnt ligands and thereby inhibits both canonical and non-canonical Wnt signaling. You may consider to include the wnt inhibitors Dickkopf and sclerostin.  

Line 321-22 “loss of endogenous Klotho in cardiomyocytes, as observed in CKD, intensifies TGFβ1 signalling and Wnt-signalling-mediated cardiac fibrosis occurrence.” What study do you refer to?

Lines 335-337. If you introduce the CKD-MBD, you should also include calcification of vessels and heart valves in the description, as soft tissue calcification is an important part of the complex disturbances. You could also just write the cardiovascular system instead of heart.

Line 391. I do disagree with the statement “Indirect lowering of FGF23 levels by reducing serum phosphate or increasing 1,25(OH2)D3 levels seems to have beneficial effect as well”. Firstly, FGF23 is not regulated by plasma phosphate on a short-term basis. It is the phosphate load that regulates FGF23. Secondly, 1,25 vit D is one of the most potent stimulator of FGF23 production in bone. Therefore, increasing plasma levels of the active vitamin D metabolite would result in an FGF23 increase instead of a decrease. Could you please comment.

Figures

The Figures are blurry and difficult to read. Can you increase the resolution?

I would suggest you remove membrane Klotho from the figures.

If you are not familiar with Wnt/β-catenin signaling, it is a bit difficult to understand the relation between the “degradation” complex and the balance between phosphorylated β-catenin and active β-catenin, in conducting β-catenin signaling.

The red lines from membrane Klotho are confusing. Also, the interaction between the signaling pathways downstream. Maybe you could give the figures a second look.

Round 2

Reviewer 1 Report

The authors have greatly improved their manuscript. I am of the opinion that it would be  worthwhile to include Klotho in the figures (if included correctly), since it is an important factor that interacts with many of the relevant pathways, as the authors describe. The figures, however, are clearer than they were and it is up to the authors and editorial team to decide how detailed they should be.